# Exploiting 3D Shape Bias towards Robust Vision

## Abstract

Robustness research in machine vision faces a challenge. Many variants of ImageNet-scale robustness benchmarks have been proposed, only to reveal that current vision systems fail under distributional shifts. Although aiming for higher robustness accuracy on these benchmarks is important, we also observe that simply using larger models and larger training datasets may not lead to true robustness, demanding further innovation. To tackle the problem from a new perspective, we encourage closer collaboration between the robustness and 3D vision communities. This proposal is inspired by human vision, which is surprisingly robust to environmental variation, including both naturally occurring disturbances and artificial corruptions. We hypothesize that such robustness, at least in part, arises from our ability to infer 3D geometry from 2D retinal projections. In this work, we take a first step toward testing this hypothesis by viewing 3D reconstruction as a pretraining method for building more robust vision systems. We introduce a novel dataset called Geon3D, which is derived from objects that emphasize variation across shape features that the human visual system is thought to be particularly sensitive. This dataset enables, for the first time, a controlled setting where we can isolate the effect of "3D shape bias" in robustifying neural networks, and informs new approaches for increasing robustness by exploiting 3D vision tasks. Using Geon3D, we find that CNNs pretrained on 3D reconstruction are more resilient to viewpoint change, rotation, and shift than regular CNNs. Further, when combined with adversarial training, 3D reconstruction pretrained models improve adversarial and common corruption robustness over vanilla adversarially-trained models. We hope that our findings and dataset will encourage exploitation of synergies between the robustness researchers, 3D computer vision community, and computational perception researchers in cognitive science, paving a way for achieving human-like robustness under complex, real-world stimuli conditions.

## 1 Introduction

Building robust vision systems is a major open problem. Tremendous efforts have been made since adversarial examples were first reported [36], and yet adversarial robustness remains perhaps the most important challenge in safe, real-world deployment of modern computer vision systems. Ensuring robustness against more common distributional shifts such as blur and snow also remains a significant challenge [18]. As clean ImageNet accuracy saturates, the research community has developed various ImageNet-scale benchmarks to evaluate the performance of vision models under distributional shifts such as broader viewpoint variability [3], style and texture change [15], geographic shifts [19]. These benchmarks, as well as the recent algorithms that are evaluated using smaller-scale datasets such as MNIST and CIFAR10 [38, 39], reveal that current vision systems have plenty of room for improvement in terms of robustness.

Submitted to 3rd Workshop on Shared Visual Representations in Human and Machine Intelligence (SVRHM 2021) of the Neural Information Processing Systems (NeurIPS) conference.

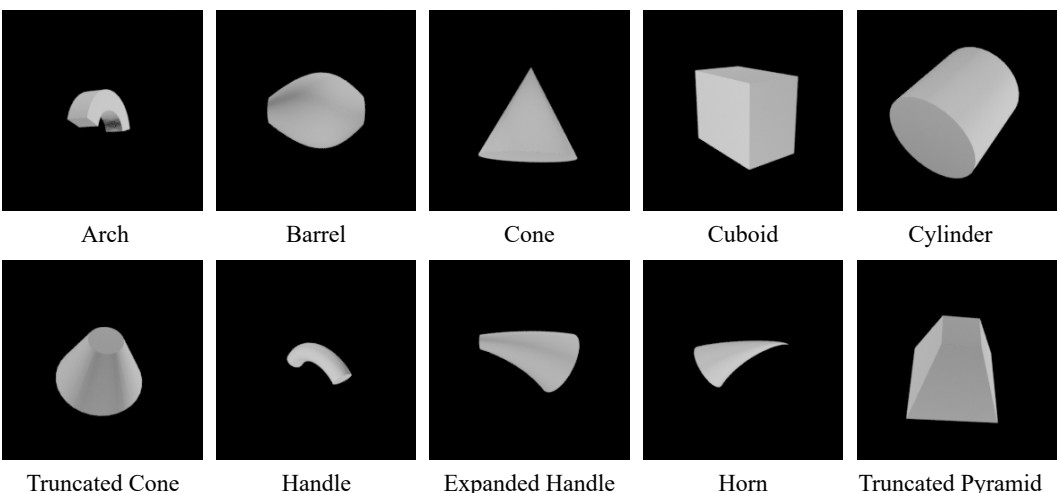

|        |        |      |        |          |
|--------|--------|------|--------|----------|
| Arch | Barrel | Cone | Cuboid | Cylinder |

| | | | | |
|--------|--------|------|--------|----------|
| Truncated Cone | Handle | Expanded Handle | Horn | Truncated Pyramid |

Figure 1: Examples of 10 Geon categories from Geon3D-10. The full list of 40 Geons we construct (Geon3D-40) is provided in the Appendix.

So far, robustness research in machine vision focuses on classification. Models trained for image classification might learn to associate class labels with a limited range of surface-related cues such as image contours, but they do not fully or explicitly reflect the relationship between 3D objects and how they are projected to images. On the contrary, the human visual system recovers rich three-dimensional (3D) geometry, including objects, shapes and surfaces, from two-dimensional (2D) retinal inputs. This ability to make inferences about the underlying scene structure from input images—also known as analysis-by-synthesis—is thought to be critical for the robustness of biological vision to occlusions, distortions, and lighting variations [41, 26].

While aiming for higher accuracy on ImageNet-scale benchmarks is important, the current landscape of robustness research shows that we face a clear challenge [37]. In fact, the consensus seems to be that large models and large training data work well for some distribution shifts, but nothing consistently help in all variants of ImageNet robustness benchmarks, awaiting methodological innovation to achieve human-level robustness [19]. To unblock the situation, we advocate closer collaboration between the robustness and 3D vision communities, in the hope of fostering new types of robustness research. This paper serves as a first step towards this effort, where we focus on learning features to facilitate inferences about 3D object shape. Our goal is to test the hypothesis that shape bias— learning representations that enable accurate inferences of 3D from 2D, which we refer to as "3D shape bias"—will induce robustness to naturally occurring challenging viewing conditions (e.g., fog, snow, brightness) and artificial image corruptions (e.g., due to adversarial attacks).

To achieve this, we introduce *Geon3D*—a novel dataset comprised of simple yet realistic shape variations, derived from the human object recognition hypothesis called Geon Theory [5]. This dataset enables us to study, in a controlled setting, 3D shape bias of 3D reconstruction models that learn to represent shapes solely from 2D supervision [28]. We find that CNNs trained for 3D reconstruction are more robust to unseen viewpoints, rotation and translation than regular CNNs. Moreover, when combined with adversarial training, 3D reconstruction pretraining improves common corruption and adversarial robustness over CNNs that only use adversarial training. These results suggest that the Geon3D dataset provides a controlled and effective measure of robustness, and unlike existing, commonly used datasets in this area such as CIFAR10 and ImageNet-C, Geon3D guides novel approaches by facilitating an interface between robust machine learning and 3D reconstruction. (Please see the Related Work section for a discussion of Geon3D in the context of existing 3D shape datasets.)

Biological vision is not only about object classification or localization, but also about making rich inference about the underlying causes of scenes such as 3D shapes and surfaces [29, 41, 26]. We hope our findings and dataset will encourage the community to tackle robustness problems through the lens of 3D inference and the perspective of perception as analysis-by-synthesis, toward the combined goals of building machine vision systems with human-like richness and reliability.

## 2 Approach

We first describe the Geon Theory, which our dataset construction relies on. Next, we explain the data generation process used in the creation of Geon3D (§2.1), and how we train a 3D reconstruction model (§2.2).

### 2.1 Geon3D Benchmark

The concept of *Geons*—or *Geometric ions*—was originally introduced by Biederman as the building block for his Recognition-by-Components (RBC) Theory [5]. The RBC theory argues that human shape perception segments an object at regions of sharp concavity, modeling an object as a composition of Geons—a subset of generalized cylinders [6]. Similar to generalized cylinders, each Geon is defined by its axis function, cross-section shape, and sweep function. In order to reduce the possible set of generalized cylinders, Biederman considered the properties of the human visual system. He noted that the human visual system is better at distinguishing between straight and curved lines than at estimating curvature; detecting parallelism than estimating the angle between lines; and distinguishing between vertex types such as an arrow, Y, and L-junction [21].

Table 1: Latent features of Geons. S: Straight, C: Curved, Co: Constant, M: Monotonic, EC: Expand and Contract, CE: Contract and Expand, T: Truncated, P: End in a point, CS: End as a curved surface

| Feature | Values |
|---|---|
| Axis | S, C |
| Cross-section | S, C |
| Sweep function | Co, M, EC, CE |
| Termination | T, P, CS |

Table 2: Similar Geon categories, where only a single feature differs out of four shape features. "T." stands for "Truncated". "E." stands for "Expanded".

| Geon Category | Difference |
|---|---|
| Cone vs. Horn | Axis |
| Handle vs. Arch | Cross-section |
| Cuboid vs. Cyllinder | Cross-section |
| T. Pyramid vs. T. Cone | Cross-section |
| Cuboid vs. Pyramid | Sweep function |
| Barrel vs. T. Cone | Sweep function |
| Horn vs. E. Handle | Termination |

Our focus in this paper is not the RBC theory or whether it is the right way to think about how we see shapes. Instead, we wish to build upon the way Biederman characterized these Geons. Biederman proposed using two to four values to characterize each feature of Geons. Namely, the axis can be straight or curved; the shape of cross section can be straight-edged or curved-edged; the sweep function can be constant, monotonically increasing / decreasing, monotonically increasing and then decreasing (i.e. expand and contract), or monotonically decreasing and then increasing (i.e. contract and expand); the termination can be truncated, end in a point, or end as a curved surface. A summary of these dimensions is given in Table 1.

Representative Geon classes are shown in Figure 1. For example, the "Arch" class is uniquely characterized by its curved axis, straight-edged cross section, constant sweep function, and truncated termination. These values of Geon features are *nonaccidental*—we can determine whether the axis is straight or curved from almost any viewpoint, except for a few *accidental* cases. For instance, an arch-like curve in the 3D space is perceived as a straight line only when the viewpoint is aligned in a way that the curvature vanishes. These properties make Geons an ideal dataset to analyze 3D shape bias and part-level robustness of vision models. For details of data preparation, see Appendix.

### 2.2 3D reconstruction as pretraining

To explore advantages of direct approaches to induce shape bias in vision models, we turn our attention to a class of 3D reconstruction models. The main hypothesis of our study is that the task of 3D reconstruction pressures the model to obtain robust representations.

Recently, there has been significant progress in learning-based approaches to 3D reconstruction, where the data representation can be classified into voxels [10, 32], point clouds [14, 1], mesh [22, 17], and neural implicit representations [25, 9, 31, 35]. We focus on neural implicit representations, where models learn to implicitly represent 3D geometry in neural network parameters after training. We avoid models that require 3D supervision such as ground truth 3D shapes. This is because we are

interested in models that only require 2D supervision for training and how inductive bias of 2D-to-3D inference achieves robustness.

Specifically, we use Differentiable Volumetric Rendering (DVR) [28], which consists of a CNN-based image encoder and a differentiable neural rendering module. We train DVR to reconstruct 3D shapes of Geon3D-10. For more details of DVR and 3D reconstruction, we refer the readers to the original paper [28].

# 3 Experimental Results

In this section, we demonstrate how 3D shape bias improves model robustness on the Geon3D-10 classification under various image perturbations. Our 3D-shape-biased classifier is based on the image encoder of the 3D reconstruction model (DVR) that is pretrained to reconstruct Geon3D-10. We add a linear classification layer on top of the image encoder, and then finetune, either just that linear layer (**DVR-Last**) or the entire encoder (**DVR**), for Geon3D-10 classification. Our baseline is a vanilla neural network (**Regular**) that is trained normally for Geon3D-10 classification. To see the difference between 3D shape bias and 2D shape bias in the sense of [15], we also evaluate the following models, which are hypothesized to rely their prediction more on shape than texture. **Stylized** is a model trained on Stylized images of Geons. **Adversarially trained network** (**AT**) is a network that uses adversarial examples during training [24]. **InfoDrop** [34] is a recently proposed model that induces 2D shape bias by decorrelating each layer's output with texture. To control for variation in network architectures, we use ImageNet-pretrained ResNet18 for all models we tested. The image encoder of DVR is also initialized using ImageNet-pretrained training for 3D reconstruction of Geons.

**Background variations**   To quantify the effect of textured background, we prepare three versions of Geon3D-10: black background, random textured background (Geon3D-10-RandTextured), and correlated background (Geon3D-10-CorrTextured). For Geon3D-10-RandTextured, we replace each black background with a random texture image out of 10 texture categories chosen from the Describable Textures Dataset (DTD) [11]. For Geon3D-10-CorrTextured, we choose 10 texture categories from DTD and introduce spurious correlations between Geon category and texture class (i.e., each Geon category is paired with one texture class). Examples of Geon3D with textured background are shown in Figure 4 (Right). These three versions of our dataset allow us to analyze more realistic image conditions as well as to test robustness despite variation and distributional shifts in textures.

**Accuracy under rotation and translation (shifting pixels)**   CNNs are known to be vulnerable to rotation and shifting of the image pixels [2]. As shown in Table 3, our model (DVR) pretrained with 3D reconstruction performs better than all other models under rotation and shift even though it is not explicitly trained to defend against those attacks. We observe that DVR-Last performs second best, indicating that this "for free" robustness to rotation and shift is largely in place even when finetuning on the classification task is restricted to only linear decoding of the categories.

Table 3: Accuracy of shape-biased classifiers against rotation and shifting of pixels on Geon3D under unseen viewpoints. We randomly add rotations of at most $30°$ and translations of at most 10% of the image size in each $x, y$ direction. We report the mean accuracy and standard deviation over 5 runs of this stochastic procedure over the entire evaluation set.

|  | REGULAR | INFODROP | STYLIZED | AT-$L_2$ | AT-$L_\infty$ | DVR-LAST | DVR |
|---|---|---|---|---|---|---|---|
| ROTATION | $82.18_{(1.06)}$ | $80.76_{(0.69)}$ | $78.47_{(0.57)}$ | $87.00_{(0.57)}$ | $89.58_{(0.48)}$ | $90.44_{(0.30)}$ | $\mathbf{93.46}_{(0.44)}$ |
| SHIFT | $72.28_{(0.43)}$ | $71.86_{(0.63)}$ | $61.44_{(0.29)}$ | $53.84_{(0.71)}$ | $61.50_{(1.11)}$ | $73.24_{(0.73)}$ | $\mathbf{76.52}_{(0.89)}$ |

## 3.1 Robustness against Common Corruptions

In this section, we show that, when combined with adversarial training, 3D pretrained models (denoted as DVR+AT-$L_2$ and DVR+AT-$L_\infty$) improve robustness against common image corruptions, above and beyond what can be accomplished just using adversarial training. For these models, we use adversarial training during the finetuning of the 3D reconstruction model for the Geon3D-10 classification task. Here we evaluate the effect of 3D shape bias not only in the somewhat sterile

scenario of the clean, black background images, but also using the background-textured versions of our dataset. To do this, we train all models using Geon3D-10-RandTextured, where we replace the black background with textures randomly sampled from DTD (see Figure 4, right panel, for examples). During evaluation, we use unseen viewpoints.

The results are shown in Table 4. We see that starting adversarial training from DVR-pretrained weights improves robustness across all corruption types, over what can be achieved by only either AT-$L_2$ or AT-$L_\infty$. DVR-AT and AT models fail on "Contrast" and "Fog". This has been a known issue for AT [16], which requires future work to explore. While Stylized performs best under certain corruption types, we can see that DVR-AT-$L_2$ leads to broader robustness across the corruptions we considered.

Table 4: Accuracy of classifiers against common corruptions under unseen viewpoints. All models are trained and evaluated on Geon3D-10 with random textured background. Pretraining on 3D shape reconstruction using DVR leads to broader robustness relative to other models.

|  | REGULAR | INFODROP | STYLIZED | AT-$L_2$ | AT-$L_\infty$ | DVR+AT-$L_2$ | DVR+AT-$L_\infty$ |
|---|---|---|---|---|---|---|---|
| INTACT | 0.741 | 0.596 | 0.701 | 0.691 | 0.464 | **0.758** | 0.513 |
| PIXELATE | 0.608 | 0.458 | 0.653 | 0.623 | 0.415 | **0.719** | 0.470 |
| DEFOCUS BLUR | 0.154 | 0.152 | 0.402 | 0.490 | 0.298 | **0.605** | 0.349 |
| GAUSSIAN NOISE | 0.222 | 0.465 | 0.601 | 0.555 | 0.412 | **0.701** | 0.470 |
| IMPULSE NOISE | 0.187 | 0.270 | 0.497 | 0.322 | 0.136 | **0.594** | 0.148 |
| FROST | 0.144 | 0.269 | **0.638** | 0.142 | 0.209 | 0.148 | 0.240 |
| FOG | 0.338 | 0.281 | **0.659** | 0.187 | 0.120 | 0.264 | 0.130 |
| ELASTIC | 0.427 | 0.314 | 0.428 | 0.416 | 0.266 | **0.499** | 0.307 |
| JPEG | 0.414 | 0.422 | 0.634 | 0.629 | 0.434 | **0.731** | 0.484 |
| CONTRAST | 0.408 | 0.286 | **0.673** | 0.141 | 0.120 | 0.179 | 0.135 |
| BRIGHTNESS | 0.525 | 0.518 | **0.702** | 0.500 | 0.388 | 0.549 | 0.429 |
| ZOOM BLUR | 0.334 | 0.238 | 0.560 | 0.518 | 0.327 | **0.639** | 0.378 |

## 3.2 3D Pretraining Improves Adversarial Robustness

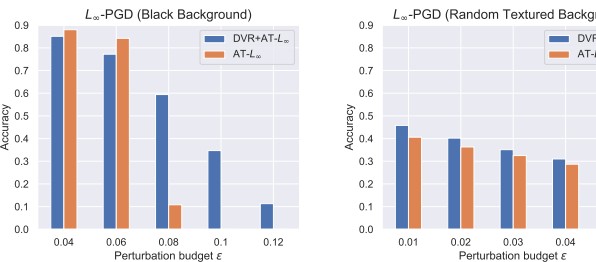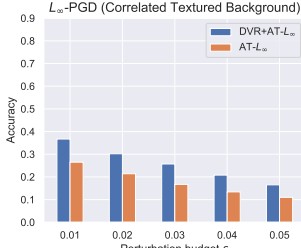

Figure 2: Robustness comparison between AT-$L_\infty$ and DVR+AT-$L_\infty$ with increasing perturbation budget $\epsilon$ on three variations of Geon3D-10. We use $L_\infty$-PGD with 100 iterations and $\epsilon/10$ to be the stepsize. See Appendix for AT-$L_2$ results, where we also find that 3D pretraining improves vanilla AT models.

In this section, we show that 3D pretrained AT models improve adversarial robustness over vanilla AT models. We attack our models using $L_\infty$-PGD [24], with 100 iterations and $\epsilon/10$ to be the stepsize, where $\epsilon$ is the perturbation budget. We compare AT-$L_\infty$ and DVR+AT-$L_\infty$ for black, randomly textured, and correlated textured backgrounds. The results are shown in Figure 2. In the black background set, while 3D pretrained AT slightly performs worse than vanilla AT for smaller epsilon values, it significantly robustifies AT-trained models for large epsilons. A small but appreciable gain in robustness can be seen for the other two backgrounds types. These pattern of results are consistent across attack types, with DVR providing significant robustness over vanilla AT under the $L_2$ regime (see Appendix).

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

# A   Additional experiments

## A.1   3D shape bias improves generalization to unseen views and reduces similar category confusion

One of the crucial but often overlooked examples of 3D shape bias that human vision has is "visual completion" [30], which refers to our ability to infer portions of surface that we cannot actually see. For instance, when we look at the top-left image in Figure 4, we automatically recognize it as a whole cube, even though we cannot see its rear side. We view the task of 3D reconstruction as a way to build such an ability into neural networks. In this section, we investigate how such 3D shape bias of DVR improves classification of similar Geon categories under unseen viewpoints, testing both DVR (where we finetune all layers of the image encoder) and DVR-Last (where we finetune only the top classification layer of the image encoder).

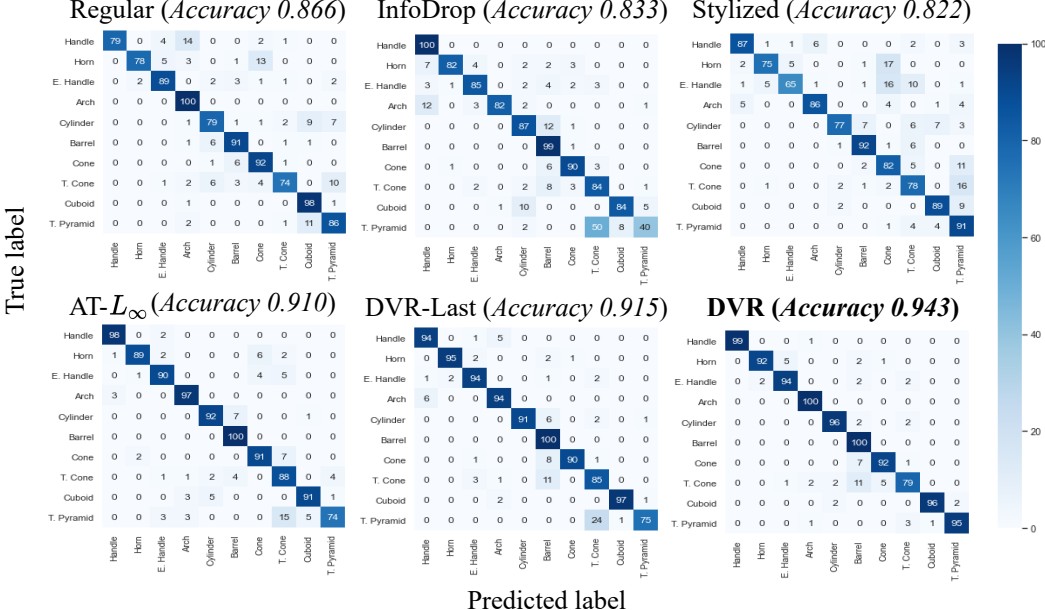

Figure 3: Accuracy per Geon category under unseen viewpoints. Even though all models perform reasonably well, there is still a range of overall accuracy values. In addition, we see that when networks make a mistake, it is often between similar Geon categories (see Table 2 for a list of similar Geon categories). Regular: a baseline model; InfoDrop: a shape-biased model; AT: adversarially trained; Stylized: a network trained on "stylized" version of Geon3D; DVR: We use pretrained weights of the image encoder of Differentiable Volumetric Rendering (3D reconstruction model), a 3D reconstruction model, and finetune all of its layers on the Geon3D-10 classification task. DVR-Last refers to the version where we finetune only the last classification layer.

The results of per-category classification are shown in Figure 3. We say two Geons are similar when there is only a single shape feature difference, as summarized in Table 2. We see that networks often misclassify similar Geon categories. The vanilla neural network (Regular) often misclassifies "Cone" vs. "Horn", "Handle" vs. "Arch", "Cuboid" vs. "Truncated pyramid", as well as "Truncated cone" vs. "Truncated pyramid".The Geon pairs the InfoDrop model misclassifies include: "Arch" vs. "Handle", "Cyllinder" vs. "Barrel", "Cuboid" vs. "Cyllinder" and "Truncated pyramid" vs. "Truncated cone", which are all pairs with single shape feature difference.

Notably, the Stylized model, which is hypothesized to increase bias towards shape-related features, makes a number of mistakes for similar Geon classes (i.e. "Horn" vs. "Cone", "Cone" vs. "Truncated pyramid", and "Truncated cone" vs. "Truncated pyramid"), similar to the Regular model. This result is consistent with the finding that the Stylized approach [15] does not necessarily induce proper shape bias [27].

AT-$L_\infty$ and DVR-Last perform better than the models listed above, yet still struggle to distinguish "Truncated Pyramid" from "Truncated Cone", where the difference is whether the cross-section is curved or straight (see Table 2). On the other hand, DVR successfully distinguishes these two categories. This shows that 3D pretraining before finetuning for the task of classification facilitates recognition of even highly similar shapes. The hardest pair for DVR is "Truncated cone" vs. "Barrel", but the errors the model make appear sensible (Figure 4, middle panel): For example, when the camera points at the smaller side of the "Truncated Cone", then there is uncertainty whether the surface extends beyond self-occlusion by contracting (which would be consistent with the "Barrel" category) or the surface ends at the point of self-occlusion (which would be consistent with the category "Truncated Cone"). Indeed, when we inspected the samples of "Truncated Cone" misclassified as "Barrel" by DVR, we found that for half of those images, the larger side of "Truncated Cone" was self-occluded. Future psychophysical work should quantitatively compare errors made by these models to human behavior.

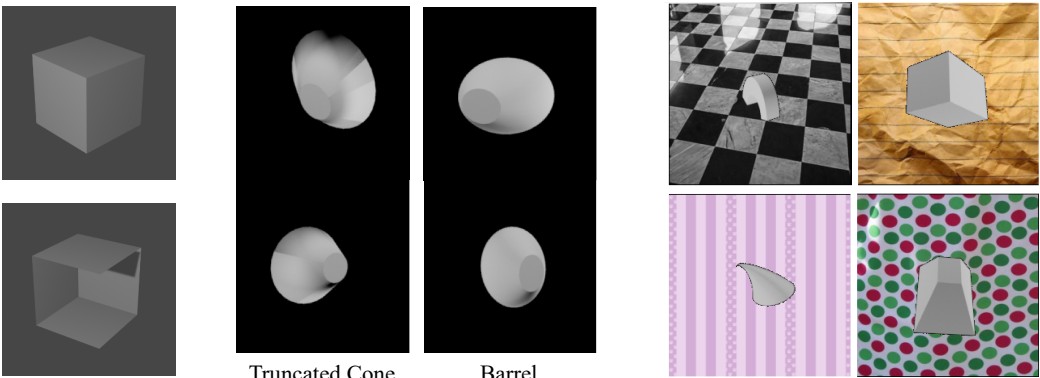

Figure 4: (Left) We humans recognize the top image as a whole cube, automatically filling in the surfaces of its rear, invisible side, although, in principle, there are infinitely many scenes consistent with the sense data , one of which is shown in the bottom image [30]. This illustrates that certain shapes are more readily perceived by the human visual system than others. (Middle) Examples of "Truncated Cone" that are misclassified as "Barrel" by DVR, next to "Barrel" exemplars shown at similar viewpoints.(Right) Example images from Geon3D-10 with textured backgrounds.

## A.2 Robustness to Distributional Shift in Backgrounds

In this section, we evaluate network's robustness to distributional shift in backgrounds. To do this, we train all the models on Geon3D-10-CorrTextured, where we introduce spurious correlation between textured background and Geon category. Therefore, during training, a model can pick up classification signal from both the shape of Geon as well as background texture. To evaluate trained models for background shift, we prepare a test set that breaks the correlation between Geon category and background texture class by cyclically shifting the texture class from $i$ to $i + 1$ for $i = 0, ..., 9$, where the class 10 is mapped to the class 0. This is inspired by [15], where they create shape-texture conflicts to measure 2D shape bias in networks trained for ImageNet classification. However, in our case, distributional shift from training to test set is designed to isolate and better measure shape bias by fully disentangling the contributions of texture and shape.

The results are shown in Table 5. We see that 2D shape biased models all perform worse than the 3D shape-biased model (DVR+AT-$L_\infty$). Combining AT with 3D pretraining improves classification accuracy more than 10 % with respect to the best performing variant of AT.

Interestingly, comparing randomized vs. correlated background experiments reveals a stark difference between the two commonly used perturbations in adversarial training ($L_2$ vs. $L_\infty$). Unlike our analysis with uncorrelated, randomized backgrounds, we find that adversarial training using $L_2$ norm completely biases the model towards texture (no apparent shape bias) when such spurious correlation between texture and shape category exists.

Table 5: Accuracy of shape-biased classifiers against distributional shift in backgrounds. Here, all models are trained on Geon3D-10-CorrTextured (with background textures correlated with shape categories) and evaluated on a test set where we break this correlation. See Appendix for results using other common corruptions, where we find DVR+AT-$L_\infty$ provides broadest robustness across the corruptions we tested.

| REGULAR | INFODROP | STYLIZED | AT-$L_2$ | AT-$L_\infty$ | DVR+AT-$L_2$ | DVR+AT-$L_\infty$ |
|---------|----------|----------|----------|---------------|--------------|-------------------|
| 0.045 | 0.121 | 0.268 | 0.015 | 0.311 | 0.219 | **0.439** |

## B   How important is 3D inference?

In this section, we investigate the importance of causal 3D inference to obtain good representations. That is, we explore the impact of having an actual rendering function constrain the representations learned by a model. Our goal in this section is not to further evaluate the robustness of these features,

but to measure the efficiency of representations learned under the constraint of a rendering function for the basic task of classification.

To isolate this effect, we compare DVR to Generative Query Networks (GQN) [13]—a scene representation model that can generate scenes from unobserved viewpoints—on novel exemplars from the Geon3D-10 dataset, but using views seen during training. The crucial difference between DVR and GQN is that GQN does not model the geometry of the object explicitly with respect to an actual rendering function. Therefore, the decoder of GQN, which is another neural network based on ConvLSTM, is expected to learn rendering-like operations solely from an objective that aims to maximize the log-likelihood of each observation given other observations of the same scene as context. To control for the difference of network architecture, we train DVR using the same image encoder architecture as GQN, since when we used ResNet18 as an image encoder, GQN did not converge.

Examples of generated images of Geons from GQN are shown in Figure 5 (Left). As we can see, GQN successfully captures the object from novel viewpoints.

To assess the power of representations learned by GQN in the same way as DVR, we take the representation network and add a linear layer on top. We then finetune the linear layer on 10-Geon classification, while freezing the rest of the weights. We compare this model to the architecture-controlled version of the DVR-Last model.

Since GQN can take more than one view of images, we prepare 6 models that are finetuned based on either of {1, 2, 4, 8, 16, 32}-views. The resulting test accuracy of finetuned GQN encoders against the number of views is shown in Figure 5 (Right). Despite the strong viewpoint generalization of GQN, we see that finetuned GQN requires more than 2 views (i.e., 3 or 4 views) to reach the DVR level accuracy, and only outperforms DVR after we feed more than 8 views. This suggests that the inductive bias from 3D inference is more efficient to obtain good representations.

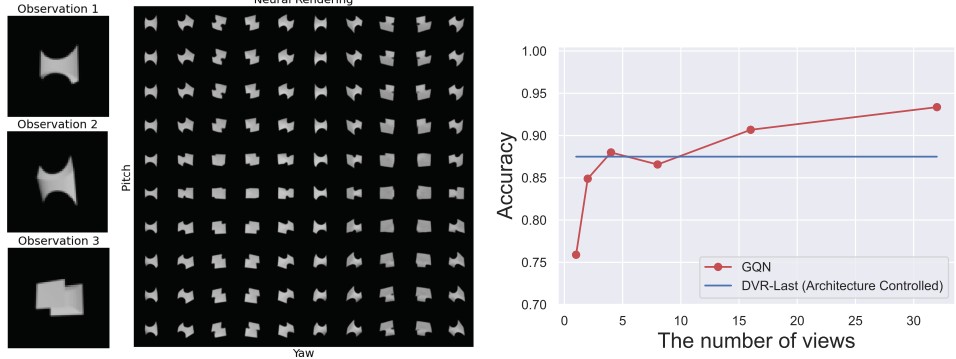

Figure 5: Left: Example Geon images rendered from GQN based on 3 views. Right: GQN Test Accuracy v.s. the number of views. As a reference, we also plot the 1-view DVR accuracy. Here, we used the same architecture for the image encoders of DVR and GQN.

### B.1 Adversarial Robustness

In Figure 6, we provide additional results for adversarial robustness, where we attack AT-$L_2$ using $L_\infty$-PGD. Similar to the case of AT-$L_\infty$, we see that 3D pretraining improves robustness over the vanilla AT models for all background settings.

### B.2 Robustness to Common Corruptions

In this section, we provide additional results for common corruptions. In Table 6, we provide the results for the black background setting. Here again we see that 3D pretraining further improves vanilla AT models. In Table 7, we provide more detailed results of distributional shift in the backgrounds. Even after adding image corruptions, we still see that DVR+AT performs best, confirming that 3D shape bias from 3D pretraining complements the performance of AT to increase model robustness.

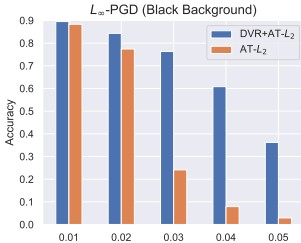 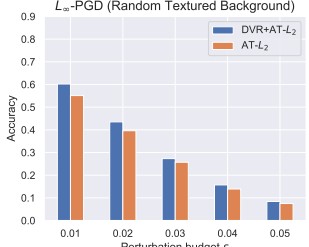 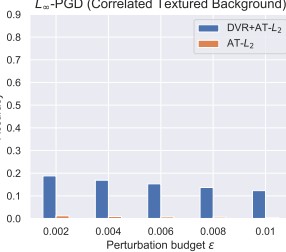

Figure 6: Robustness comparison between AT-$L_2$ and DVR+AT-$L_2$ with increasing perturbation budget $\epsilon$ on three variations of Geon3D-10. We attack our models using $L_\infty$-PGD with 100 iterations and $\epsilon/10$ to be the stepsize.

Table 6: Accuracy of shape-biased classifiers against common corruptions under unseen views on Geon3D-10 (black backgrounds).

|  | REGULAR | INFODROP | STYLIZED | AT-$L_2$ | AT$L_\infty$ | DVR+AT-$L_2$ | DVR+AT-$L_\infty$ |
|---|---|---|---|---|---|---|---|
| INTACT | 0.866 | 0.845 | 0.822 | 0.908 | 0.910 | 0.912 | **0.92** |
| PIXELATE | 0.685 | 0.773 | 0.781 | 0.905 | 0.910 | 0.911 | **0.919** |
| DEFOCUS BLUR | 0.303 | 0.247 | 0.755 | 0.900 | **0.909** | 0.897 | **0.909** |
| GAUSSIAN NOISE | 0.548 | 0.291 | 0.803 | 0.620 | 0.885 | 0.914 | **0.919** |
| IMPULSE NOISE | 0.140 | 0.190 | 0.750 | 0.542 | 0.100 | 0.916 | **0.918** |
| FROST | 0.151 | 0.323 | **0.783** | 0.140 | 0.100 | 0.22 | 0.3 |
| FOG | 0.138 | 0.163 | **0.764** | 0.100 | 0.100 | 0.119 | 0.149 |
| ELASTIC | 0.612 | 0.635 | 0.617 | 0.628 | 0.664 | 0.645 | **0.655** |
| JPEG | 0.799 | 0.821 | 0.810 | 0.905 | 0.911 | 0.912 | **0.92** |
| CONTRAST | 0.510 | 0.180 | **0.772** | 0.163 | 0.258 | 0.213 | 0.335 |
| BRIGHTNESS | 0.552 | 0.832 | 0.818 | 0.160 | 0.137 | 0.385 | **0.931** |
| ZOOM BLUR | 0.475 | 0.462 | 0.748 | 0.891 | 0.917 | 0.902 | **0.92** |

## C  Related Work and Discussions

**3D datasets**. Geon3D is smaller in scale and less complex in shape variation relative to some of the existing 3D model datasets, including ShapeNet [8] and ModelNet [43]. These datasets have been instrumental for recent advances in 3D computer vision models (e.g. Niemeyer et al. [28], Sitzmann et al. [35]). However, at a practical level, these 3D model datasets are not yet suitable for our goal (which is to establish whether introducing 3D shape bias into vision models induce robustness): Even though existing learning-based 3D reconstruction models can perform well when trained on a single or a very small number of categories from these datasets, these models do not scale well with increasing number of object categories. For example, on ShapeNet, when these models are required to learn a non-trivial number of object categories (e.g., 10 or more) at the same time, the resulting 3D shape reconstructions degrade significantly, unable to capture many salient aspects of shape variation across and within categories. For us, such failure confounds inferences we can make about the role of shape bias in robustness, which is our central question: Would a negative result be because the model does not perform well on the reconstruction task to begin with or is it that shape bias has no benefit for robustness? We deliberately designed Geon3D to allow us to take advantage of the state-of-the-art in learning-based 3D reconstruction models (in this work, the DVR model): It provides a non-trivial number of distinct shape categories, with considerable shape variation within and across categories, yet remain tractable to learn by these existing models. As we demonstrate in this work, despite its simplicity relative to these larger datasets, Geon3D reveals that the current vision models struggle with image corruptions and that 3D shape bias induces robustness. Our results based on Geon3D provide compelling evidence that to achieve robustness against distributional shifts and adversarial examples, a promising and effective approach is to build models with 3D shape bias. In future work, we are excited to explore this hypothesis in the context of more complex shapes and real-world objects and scenes.

**Analysis-by-synthesis**. Our proposal of using 3D inference to achieve robust vision shares the same goal as analysis-by-synthesis [23, 41, 40]. In DVR, we can see its encoder as a recognition

Table 7: Accuracy of shape-biased classifiers against common corruptions under unseen views on Geon3D-10 with textured background swap.

|  | REGULAR | INFODROP | STYLIZED | AT-$L_2$ | AT-$L_\infty$ | DVR+AT-$L_2$ | DVR+AT-$L_\infty$ |
|---|---|---|---|---|---|---|---|
| INTACT | 0.045 | 0.121 | 0.268 | 0.015 | 0.311 | 0.219 | **0.439** |
| PIXELATE | 0.044 | 0.096 | 0.275 | 0.017 | 0.306 | 0.201 | **0.415** |
| DEFOCUS BLUR | 0.044 | 0.093 | 0.268 | 0.024 | 0.242 | 0.206 | **0.338** |
| GAUSSIAN NOISE | 0.046 | 0.160 | 0.269 | 0.015 | 0.320 | 0.209 | **0.408** |
| IMPULSE NOISE | 0.058 | 0.096 | **0.228** | 0.015 | 0.078 | 0.207 | 0.147 |
| FROST | 0.020 | 0.138 | **0.255** | 0.070 | 0.149 | 0.144 | 0.227 |
| FOG | 0.032 | 0.114 | **0.273** | 0.077 | 0.099 | 0.149 | 0.124 |
| ELASTIC | 0.044 | 0.109 | 0.260 | 0.100 | 0.196 | 0.176 | **0.264** |
| JPEG | 0.041 | 0.089 | 0.264 | 0.016 | 0.306 | 0.206 | **0.419** |
| CONTRAST | 0.055 | 0.107 | **0.274** | 0.066 | 0.090 | 0.148 | 0.126 |
| BRIGHTNESS | 0.036 | 0.127 | 0.268 | 0.026 | 0.270 | 0.189 | **0.379** |
| ZOOM BLUR | 0.081 | 0.082 | 0.290 | 0.032 | 0.269 | 0.249 | **0.375** |

network [12], mapping 2D images to their underlying shape, appearance, and pose parameters under a structured generative model based on a neural rendering function. Even though previous work considered adversarial robustness of variational autoencoders [33], our study is first to evaluate robustness arising from analysis-by-synthesis type computations under 3D scenes.

# D   Datasheet

A line of work in psychophysics of human visual cognition have argued that the visual system exploits certain types of shape features in inferring 3D structure and geometry. In Geon3D, by treating these shape features as the dimensions of variation, we model 40 classes of 3D objects, and render them from random viewpoints, resulting in an image set and their corresponding camera matrices.

**Data Preparation**   We construct each Geon using Blender —an open-source 3D computer graphics software [7].

An advantage of Geons over other geometric primitives such as superquadrics [4] is that the shape categorization of Geons is qualitative rather than quantitative. Thus, each Geon category affords a high degree of in-class shape deformation, as long as the four defining features of each shape class remains the same. Such flexibility allows us to construct a number of different 3D model instances for each Geon class by expanding or shrinking the object along the x, y, or z-axis. For each axis, we evenly sample the 11 scaling parameters from the interval [0.5, ..., 1.5] with a step size 0.1, resulting in 1331 3D model instances for each Geon category.

**Rendering and data splits**   We randomly sample 50 camera positions from a sphere with the object at the origin. For each model instance, 50 images are rendered using these camera positions with resolution of 224x224. We then split the data into train/validation/test with ratio 8:1:1 using model instance ids, where each instance id corresponds to the scaling parameters described above. We also make sure that all Geon categories are uniformly sampled in each of train/validation/test sets.

**Dataset distribution**   The full Geon3D-40 (black background) will be available for download after publication. Geon3D is distributed under the CC BY-SA 4.0 license.[1] We plan to maintain different versions of Geon3D as we extend the dataset to include more complicated objects by combining Geon3D as parts. The authors bear all responsibility in case of violation of rights and confirmation of the data license. Upon publication, the dataset website will become available, where we will add structured metadata to a dataset's meta-data page, a persistent dereferenceable identifier, and any future updates.

**How to use Geon3D**   Our dataset contains 40 Geon categories, where each folder contains 1331 subfolders. The name of the subfolder represents the scaling factors for the x, y, and z direction. For

---

[1] https://creativecommons.org/licenses/by-sa/4.0/legalcode

example, `0.5_1.0_1.3` means the Geon model is scaled by 0.5, 1.1, and 1.3 for x, y, and z axis, respectively. Each subfolder contains the 'rgb' folder, 'mask' folder, and 'pose' folder. The 'rgb' folder contains 50 images taken from 50 random viewpoints. The 'mask' and 'pose' folders are used for 3D reconstruction tasks. An example code will be provided to demonstrate how to load these 'mask' and 'pose' information to do 3D reconstruction task.

**Benchmarking metric** Our metric for benchmarking model robustness is accuracy under different noise types (e.g. Section 3.1, 3.2, 3.3, 3.4). Unless we achieve near-perfect accuracy on each noise type, we don't think robustness issues are solved on this dataset. We would like to avoid using a single metric such as the mean robust accuracy, since such a metric inevitably obscures the intricate differences that arise from different noise types.

**List of 40 Geons** In Figure 7, we provide a list of 40 Geons we have constructed. The label for each Geon class represents the four defining shape features, in the order of "axis", "cross section", "sweep function", "termination", as described in the main paper. We put "na" for the termination when the sweep function is constant. We also distinguish the two termination types "c-inc" and "c-dec" when the sweep function is monotonic. For instance, "c-inc" means that the curved surface is at the end of the increasing sweep function, whereas "c-dec" means that the curved surface is at the end of the decreasing sweep function. As a reference, here is the mapping between the name and the code of 10 Geons we used in 10-Geon classification: "Arch": `c_s_c_na`, "Barrel": `s_c_ec_t`, "Cone": `s_c_m_p`, "Cuboid": `s_s_c_na`, "Cylinder": `s_c_c_na`, "Truncated cone": `s_c_m_t`, "Handle": `c_c_c_na`, "Expanded Handle": `c_c_m_t`, "Horn": `c_c_m_p`, "Truncated pyramid": `s_s_m_t`.

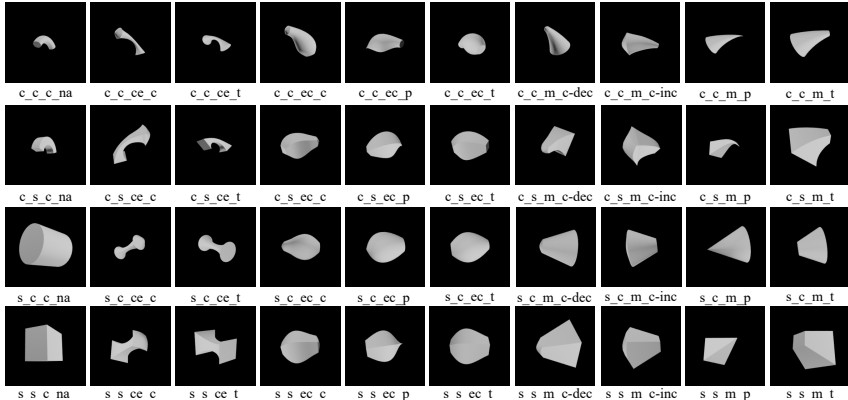

Figure 7: The list of 40 Geons we constructed.

# E   Reproducibility: Training details

We used GeForce RTX 2080Ti GPUs for all of our experiments. GQN training takes about a week until convergence on a single GPU. DVR 3D reconstruction training takes roughly about 1.5 days on a single GPU. The hyperparameters for 10-Geon classification, described in the main paper, were chosen by monitoring the model convergence on the validation set. The inputs to all models during classification are only RGB images. (Camera matrices are only used for the rendering module during pretraining for 3D reconstruction.)

**DVR** We used the code [2] open-sourced by Niemeyer et al. [28]. We followed the default hyperparameters recommended by Niemeyer et al. [28] for 3D reconstruction training, with the exception of batch size, which we set 32 to fit into a single GPU memory.

**Adversarial Training** Through extensive experiments, Zhang & Zhu [42] demonstrate that AT models develop 2D shape bias, which is considered to explain, in part, the strong adversarial

---

[2] `https://github.com/autonomousvision/differentiable_volumetric_rendering`

robustness of AT models. In our experiments, we use $L_\infty$ and $L_2$ based adversarial training. We used the python package [3] to perform adversarial training. For AT($L_2$), we use attack steps 7, epsilon 3.0, attack lr 0.5. For AT($L_\infty$), we use attack steps 7, epsilon 0.05, attack lr 0.01. use best (final) PGD step as example. Both models trained for 70 epochs with batch size 100, which was sufficient for model convergence.

**GQN**   We used the open-source code [4] to implement our GQN. Due to the training instability, we rescale the image size from 224 x 224 to 64 x 64.

**InfoDrop**   We used the original author's implementation [5]. The method exploits the fact that texture often repeats itself, and hence is highly correlated with and can be predicted by the texture information in the neighboring regions, whereas shape-related features such as edges and contours are less coupled at the locality of neighboring regions.

**Stylized**   We follow the same protocol as [15] by replacing the texture of each image of Geon3D-10 by a randomly selected texture from paintings through the AdaIn style-transfer algorithm [20]. To stylize Geon3D, we used the code [6] introduced by the original author of Stylized-ImageNet [15].

**Dataset**   For training Geon3D image classifiers, we center and re-scale the color values of Geon3D with $\mu = [0.485, 0.456, 0.406]$ and $\sigma = [0.229, 0.224, 0.225]$, which is estimated from ImageNet. We construct the 40 3D model instances as well as the whole training data in Blender. We then normalize the object bounding box to a unit cube, which is represented as `1.0_1.0_1.0` in the dataset folder.

**Background textures**   We used the following label-to-texture class mapping: {0: 'zigzagged', 1: 'banded', 2: 'wrinkled', 3: 'striped', 4: 'grid', 5: 'polka-dotted', 6: 'chequered', 7: 'blotchy', 8: 'lacelike', 9: 'crystalline' }. For the distributional shift experiment we used the following mapping: { 0: 'crystalline', 1: 'zigzagged', 2: 'banded', 3: 'wrinkled', 4: 'striped', 5: 'grid', 6: 'polka-dotted', 7: 'chequered', 8: 'blotchy', 9: 'lacelike', }. The DTD data is licensed under the Creative Commons Attribution 4.0 License. [7]

**Evaluation set**   For all the evaluation sets in the experiment section, we used the same subset of the test split, where we randomly pick 1000 model instance ids, and randomly sample 1 view out of 50 views for every model instance.

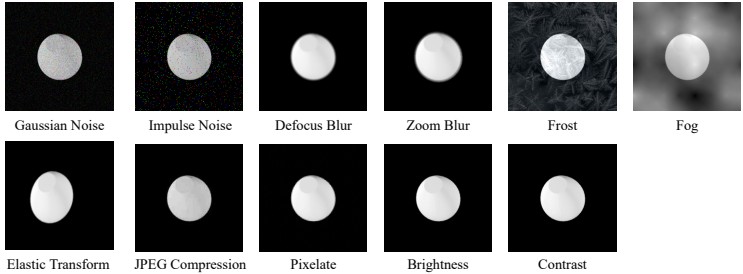

Figure 8: Examples of image corruptions.

We use the original author's code [8] to generate common corruptions shown in Figure 8.

---

[3] https://github.com/MadryLab/robustness

[4] https://github.com/iShohei220/torch-gqn

[5] https://github.com/bfshi/InfoDrop

[6] https://github.com/bethgelab/stylize-datasets

[7] https://creativecommons.org/licenses/by/4.0/, https://www.tensorflow.org/datasets/catalog/dtd

[8] https://github.com/hendrycks/robustness

