# OpenReview forum: "Exploiting 3D Shape Bias towards Robust Vision"
_NeurIPS.cc/2021/Workshop/SVRHM — SVRHM 2021 Poster_

### Official Review · Reviewer_Hgay · 2021-10-22
**Interesting ansatz with slightly unclear conclusion**

**Rating:** 6
**Confidence:** 3

**Review:**

This paper explores the effect of pretraining on 3D image reconstruction can help to increase robustness against distribution shifts. To this end, this work introduces a well controlled dataset of 3D shapes and compares against a wider array of shape-biased representation learning methods.

Overall, the manuscript is well written and easy to follow, and it explores a very interesting and timely topic. The biggest problem I see right now is that the dataset is very small (the main experiments use 10 classes with 50 images (views of the same object from different angles)). The training dataset has just 400 images with relatively low variability. This seems to result in quite varying vanilla accuracy on the test set: looking at the first row of Table 4 ("intact") suggests that baseline performance varies from 0.46 for adversarial training to 0.76 for DVR (the proposed method). The good performance of DVR is unsurprising given that it's inductive bias is straight on point and extremely important for such a small dataset. As a result I am not sure whether we are truly measuring differences in OOD robustness between the methods or rather different baseline accuracies / learning efficiencies (or a mixture of the two).

Also, it looks as if there are some inconsistencies in the numbers because the values in Table 3 are higher than in the first row of Table 4. I am most likely missing something here, but would be great to clarify. Please also clearly report the vanilla IID accuracy of all models.

Is it correct that the generator in DVR is thrown away after training?

For adversarial robustness, I am not sure the difference to AT is really significant. Could you please add a check to see if the values already converged (i.e. what happens if you increase the number of PGD steps by 5?) and maybe use another method like B&B [1] as a sanity check.

[1] https://arxiv.org/abs/1907.01003

---

### Official Review · Reviewer_DkvD · 2021-10-23
**Interesting idea but major quality and clarity issues**

**Rating:** 3
**Confidence:** 4

**Review:**

**Summary**

To improve the robustness of CNN image classifiers to various types of noise, this work proposes 3D shape reconstruction pretraining using DVR, followed by fine-tuning to improve classification robustness. The authors compare to 3 other methods for improving robustness: Stylized, AT, and InfoDrop.

**Originality**

To my knowledge the application of 3D shape pretraining using DVR for improving image classifier robustness is novel
Since the related work was not discussed in the main text, it makes it difficult for readers to assess the originality of the work

**Quality**

This paper has several quality issues
- The ending is quite abrupt. This paper needs a discussion section that identifies strengths, limitations, and future work
- Related work should be discussed in the main text
- Table 3: An important baseline missing here is doing 2D rotation and shift data augmentations (explicitly training to defend against - these “attacks”).
- Table 3: what are the details for the training process? how are viewpoints sampled during training? Could it be that training viewpoints-  are very close to viewpoint used at test time?
- L65: Need citations for CIFAR10 and ImageNet-C!
- No discussion of why the accuracies on the Geon3D-10 dataset appear so low
- No discussion of the drawbacks of using the proposed pretraining. What is the computational cost? Do the robustness gains justify the - cost?
- L60: “We find that CNNs trained for 3D reconstruction are more robust to unseen viewpoints, rotation and translation than regular - CNNs “
- how was the train/test split decided?

**Clarity**

This writing in this paper needs to be improved to enhance clarity for the reader. Below are several instances I would like to bring to the authors’ attention.

Major

- L47: what is the challenge?
- how was the train/test split decided?
- Table 3
- What is the accuracy referring to here? Train or test? Geon3D-10 or Geon3D?
- L116-117: Referring readers to the DVR paper without any explanation of the method greatly worsens the clarity of this paper. You must - provide a sufficient summary of the method since your entire paper depends on it.
- How is the task of 3d reconstruction defined?
- Table 4
- What is the accuracy referring to here? Train or test?
- The first column should be labelled here
- Does intact mean the images are left alone?
- Caption should explain each type of noise in the first column
- Caption should refer to the dataset as Geon3D-10-RandTextured to consistent
- No explanation of difference between AT-L2 and AT-Linf
- L160: Do you have an explanation for why they fail on contrast and fog?
- If DVR is pretrained on 3d shape reconstruction task, then trained to classify images, did you control for the number of epochs? Or were all models trained until convergence?

Minor

- L165: Weird spacing issue
- Nitpick: Table 3 std should be +- instead of ()
- L67: use a proper reference here
- typo and awkward first abstract sentences
- L116: Geon3D-10 is mentioned for the first time without any explanation that it is a 10-class dataset of geons (I assume)
- L142: I believe “Accuracy under rotation and translation (shifting pixels) ” should be its own subsection
- L102 needs a proper reference to appendix section
- L38: Citations would be nice here
- L130: “To control for variation in network architectures, we use ImageNet-pretrained ResNet18 for all models we tested. “
- clarification: does this apply only to InfoDrop or all models including DVR?

**Significance**

The proposed method is better at improving robustness on an image classification task compared to Stylized, AT, and InfoDrop. However, the widely used technique of data augmentations was not included as a baseline. It is unclear how this work could be applied to a more common image classification task, which limits the significance of this work.

---

### Official Review · Reviewer_b2KN · 2021-10-30
**Important question, well designed experiments, results which should be shared with community ---> Strong accept for the workshop**

**Rating:** 8
**Confidence:** 4

**Review:**

Paper Summary: The paper proposes asks the question if the ability to reconstruct 3D information from 2D scenes can help make networks more robust to transformations including changes in viewpoint, rotations and shifts, and even image corruptions. For this, the paper begins by presenting a dataset of rendered 3D geons and by pre-training using 3D reconstruction on this dataset. The results show that this hypothesis is indeed true, as such pre-training leads to more robust computer vision models.

Strengths:

1. Results clearly show that the relationship between 3D vs 2D vision tasks, and the robustness of representations learned is an interesting problem. This is extremely fascinating, and I believe that it should be shared with other researchers in the field so this relationship can be fleshed out.

2. The question is an extremely important one: robustness to common corruptions, viewpoint changes, rotations and shifts have huge ramifications both theoretically and in practice in the real world. Thus, advances made on this front are of great significance.

3. The paper is well written, easy to read and follow. The tables/figures and captions capture they high level ideas well.

Weaknesses:

Missing literature: The paper is missing two related threads of existing work which are closely related and should be added.

1. Recent work on brittleness to viewpoint changes:-

Hsueh-Ti Derek Liu, Michael Tao, Chun-Liang Li, Derek Nowrouzezahrai, and Alec Jacobson. Beyond pixel norm-balls: Parametric adversaries using an analytically differentiable renderer. In Proceedings of the International Conference on Learning Representations (ICLR), 2019.

Xiaohui Zeng, Chenxi Liu, Yu-Siang Wang, Weichao Qiu, Lingxi Xie, Yu-Wing Tai, Chi-Keung Tang, and Alan L Yuille. Adversarial attacks beyond the image space. In Proceedings of the IEEE/CVF Conference on Computer Vision and Pattern Recognition (CVPR), pages 4302–4311, 2019.

2. Recent work on generalization/robustness to viewpoint variations:-

Madan, S., Henry, T., Dozier, J., Ho, H., Bhandari, N., Sasaki, T., Durand, F., Pfister, H. and Boix, X., 2020. On the capability of neural networks to generalize to unseen category-pose combinations. arXiv preprint arXiv:2007.08032.

I would also advise the authors to cross-reference literature cited in these papers, and cite any relevant papers to place their work it in the context of existing robustness literature.

Review Summary: The question is interesting, the experiments are solid, and the paper is well written. The contributions are appropriate for a workshop paper and interesting to be shared with others working on robustness.

---

### Official Review · Reviewer_eVtD · 2021-10-31
**Handy dataset and sensible idea**

**Rating:** 8
**Confidence:** 4

**Review:**

The authors begin with the intuition that one of the things that makes human vision robust to image perturbations is our ability to see beyond 2D image data, to infer underlying 3D objects in the world. They ask whether a deep neural network explicitly trained to infer 3D shape from 2D images would similarly benefit. To test this, they create a new dataset tailored towards relatively easy 3D shape inference, by rendering "Geons" - the geometrically simple shapes proposed by Biederman et al some decades ago as the constituent elements of human 3D vision (the current paper doesn't commit itself to Biederman's view, but just uses the 40 Geon shapes as a handy library of diverse 3D shapes). The dataset is rendered with various background texture conditions (black, random texture, texture correlated with shape class). The target model (a Differentiable Volumetric Rendering network) and several alternatives from the robustness literature are trained to classify Geons in this dataset, and then tested under various image perturbation conditions.

The DVR-based 3D inference model is substantially better able to classify shapes when shown viewpoints that are rotated or shifted relative to those in its training dataset, compared to other models. When combined with adversarial training, the DVR model is also more robust to most 2D image perturbations (noise, blur, etc) than other models. Presumably this robustness does not come "for free" as the viewpoint generalisation did, since no results are shown for DVR alone (without adversarial training) here. The paper presents a solid first step towards demonstrating the usefulness of 3D inference in building robust visual representations. Ideally one would want to see evidence that pretraining on 3D inference can provide benefits on naturalistic object recognition datasets, but this is understandably a much harder challenge.

The Geon3D dataset is released for others to use, and may prove to be a handy benchmark for work involving 3D inference and view-generalisation. The paper is clearly written and the rationale, methods, and results are all well explained (given the page limitations). Figures are used effectively, and are clearly labelled. Figure and table captions are informative.

---

### Decision · Program_Chairs · 2021-11-02

Accept (Poster)